# Gastric Emptying of New-World Milk Containing A1 and A2 Β-Casein Is More Rapid as Compared to Milk Containing Only A2 Β-Casein in Lactose Maldigesters: A Randomized, Cross-Over Trial Using Magnetic Resonance Imaging

**DOI:** 10.3390/nu15040801

**Published:** 2023-02-04

**Authors:** Monica Ramakrishnan, Xiaopeng Zhou, Ulrike Dydak, Dennis A. Savaiano

**Affiliations:** 1Department of Nutrition Science, College of Health and Human Sciences, Purdue University, West Lafayette, IN 47907, USA; 2School of Health Sciences, Purdue University, West Lafayette, IN 47907, USA

**Keywords:** gastric transit, A1 β-casein, A2 β-casein, A2 milk, A1 milk, lactose maldigestion, magnetic resonance imaging, betacasomorphin-7

## Abstract

Lactose maldigesters report an increase in abdominal pain due to the consumption of milk containing a mixture of A1 and A2 β-casein as compared to milk containing only A2 β-casein. Gastric transit affects gastrointestinal symptoms and rapid transit has been associated with an increase in abdominal pain. We conducted a double-blinded, randomized, crossover trial in 10 lactose maldigesters. Subjects consumed each of the two types of milk: conventional milk containing 75% A1 β-casein and 25% A2 β-casein and A2 milk containing 100% A2 β-casein. Magnetic resonance images were acquired, and abdominal pain was rated and recorded at 0, 10, 30, 60 and 120 min after milk consumption. The volume of milk in the stomach was calculated using FSL software. The volume of milk in the stomach after consuming milk with 75% A1 β-casein and 25% A2 β-casein was significantly lower at 30 (*p* = 0.01), 60 (*p* = 0.002) and 120 (*p* < 0.001) minutes as compared to milk with 100% A2 β-casein in the 10 lactose maldigesters. The transit of New-World milk containing A1 and A2 β-casein was more rapid as compared to Old-World milk containing only A2 β-casein. This difference in transit may mediate symptoms of lactose intolerance.

## 1. Introduction

Casein and whey are the two major proteins in cow’s milk. β-casein forms approximately 30% of protein in bovine milk [1,2]. There are thirteen genetic variants of β-casein. A1 and A2 β-casein have been studied most extensively for their effects on health [3]. A2 β-casein can be found in pure bred ancestral cattle milk [4,5]. A2 β-casein, also known as the original β-casein variant, is a 209 amino-acid long protein with proline at its 67th position [6]. A1 β-casein is more common in modern European cattle milk and arose due to a single point mutation at position 67, which substituted proline with histidine [3,6]. This mutation occurred ten thousand years ago [1]. Therefore, we refer to milk containing only A2 β-casein as Old-World milk and milk containing any A1 β-casein as New-World milk. The single point mutation results in the formation of a pro-inflammatory peptide, beta-casomorphin-7, from A1 β-casein during gastric digestion [7,8].

Several studies have shown that the consumption of milk containing A1 β-casein aggravates symptoms of lactose intolerance [9,10,11,12]. Our group previously conducted a double-blinded randomized study in lactose maldigesters and lactose intolerant subjects to study the effects of milk with varying compositions of A1 and A2 β-casein on intolerance and maldigestion. We found that significantly higher abdominal pain was associated with the consumption of milk containing 75% A1 β-casein and 25% A2 β-casein as compared to milk containing 100% A2 β-casein [13]. These results were consistent with others [9,11,14].

Gastric transit time is associated with gastrointestinal symptoms [15], and rapid gastric emptying is associated with abdominal pain [16]. Magnetic Resonance Imaging [MRI] has been used as a reliable non-invasive technique to observe gastric emptying for almost two decades [17,18]. Therefore, we conducted a double-blinded randomized trial to examine the difference in the gastric emptying of kinds of milk with different A1 and A2 β-casein compositions: conventional milk (milk containing 75% A1 β-casein and 25% A2 β-casein) and A2 milk (milk containing no A1 β-casein and 100% A2 β-casein) using MRI in lactose maldigesters.

We hypothesized that the lower symptom scores for abdominal pain due to consumption of milk containing only A2 β-casein as compared to milk containing A1 and A2 β-casein was related to the difference in the gastric emptying time. 

## 2. Materials and Methods

### 2.1. Subject Enrolment, Allocation and Inclusion Criteria

This was a randomized, double-blinded, crossover trial. Subjects aged 18–65 years were recruited using flyers, advertisements in local and campus newspapers and email. The recruitment started in March 2021 and the data collection was completed in October 2022. Sixty-one people indicated an interest in the study and contacted the study staff through email or phone (Figure 1). An informed consent with a detailed study procedure and eligibility criteria was sent for review to interested subjects. Twenty-eight people were phone screened with queries regarding medication use, demographic information and body mass index (BMI) and were then classified as eligible or ineligible (Appendix A). Eligible subjects were assigned an identification number, provided written informed consent and were enrolled. Research Randomizer (randomizer.com, (accessed on 28 June 2021) was used to generate the randomization. Thirty sets of numbers (from 1–30) were randomly assigned in an order of 1 (conventional milk) and 2 (A2 milk). Eligible subjects were sequentially assigned to the numbers. Staff performed the randomization, enrolled subjects and assigned the randomized intervention to participants. The randomized milk type was sent to the clinical research center (CRC) kitchen staff via email and milk was poured into containers labeled with the subject ID. The milk type was not indicated in the containers. Study staff who assessed the outcomes and study participants were blinded to treatment. Subjects identified themselves as milk intolerant or milk avoiders. Lactose maldigesters were identified with a 20 ppm increase in hydrogen between any two time points during a 6-h hydrogen breath test (HBT) after a milk challenge dose of 0.5 g of lactose/kg body weight [19]. 

### 2.2. Exclusion Criteria

Subjects were excluded from the study if they met one or more of the following criteria: currently pregnant, abnormal gastrointestinal motility, history of gastrointestinal tract surgery, presence of a medical condition that could confound collection of adverse events, ulcers, diabetes mellitus, congestive heart failure, HIV, hepatitis B, hepatitis C, use of systemic antibiotics thirty days prior to screening; had certain implants, metallic objects or irremovable jewelry on the body that could produce artefacts or potentially harm the subjects during the MRI scan.

### 2.3. Intervention

Two milks were fed: Conventional milk containing 75% A1 β-casein and 25% A2 β-casein (Kroger^®^ 2% reduced fat; The Kroger Co., Indianapolis, IN, USA) and A2 milk containing no A1 β-casein and 100% A2 β-casein (The a2 Milk Company, Boulder, CO, USA). Both milk types were available for purchase at Payless (West Lafayette, IN, USA), Fresh thyme (West Lafayette, IN, USA) or Meijer (West Lafayette, IN, USA).

### 2.4. Nutrient Analysis

Total sugar, fat and protein in conventional milk and A2 milk were analyzed by Eurofins Food Integrity and Innovation (Eurofins Food Chemistry Testing US, Inc., Madison WI, USA). A1 and A2 β-casein in the two types of milk were analyzed using mass spectrometry at Purdue Proteomics Facility. 

#### 2.4.1. Analysis of A1 β-Casein/A2 β-Casein Using Liquid Chromatograph Triple Quadrupole Mass Spectrometer (LC-MS/MS)

One hundred microliters (100 µL) of each milk sample (conventional milk and A2 milk) were denatured using 400 µL of 8 M urea and 10 mM DTT (dithiotheitol), and vortexed for 15 min at room temperature. Denaturation with urea aids in the separation of fat from protein. Fat removal was followed by the process of centrifugation. The proteins extracted from the solution were alkylated with 20 mM IAA (iodoacetamide) and digested with pepsin for one hour at room temperature using a 1:20 enzyme-to-substrate ratio. After digestion, peptides were desalted using C_18_ Silica MicroSpin Columns (The Nest Group Inc., Southborough, MA, USA). The desalted peptides were analyzed in the Dionex UltiMate 3000 RSLC nano System combined with the Q-Exactive High-Field (HF) Hybrid Quadrupole Orbitrap MS (Thermo Fisher Scientific, Waltham, MA, USA). Peptides were then re-suspended in 3% acetonitrile (CAN)/0.1% Formic Acid (FA)/96.9% MilliQ water and 5 μL (1 μg). They were then used for liquid chromatography (LC)-MS/MS analysis. Peptides were separated using a trap (300 μm internal diameter [ID] × 5 mm packed with 5 μm 100 Å PepMap C18 medium) and the analytical columns (75 μm ID × 15 cm long packed with 3 μm of 100 Å PepMap C18 medium) (Thermo Fisher Scientific, Waltham, MA, USA) using a 120 min gradient method at a flow rate of 300 nL/min. There were two mobile phases: mobile phase A comprised 0.1% FA in water and mobile phase B consisted of 0.1% FA in 80% ACN. The linear gradient progressed from 5% B to 30% B in 80 min, 45% B in 91 min, and 100% B in 93 min. Following this, the column was held at 100% B for the next 5 min before it was brought back to 5% B and held for 20 min to equilibrate the column. The column temperature was maintained at 37 °C. MS data were acquired using a top 20 data-dependent MS/MS scan method with a maximum injection time of 100 ms and a resolution of 120,000 at 200 *m*/*z*. Fragmentation of precursor ions was performed using high-energy C-trap dissociation (HCD) with the normalized collision energy of 27 eV. MS/MS scans were acquired at a resolution of 15,000 at *m*/*z* 200. The dynamic exclusion was set at 20 s to avoid repeated scanning of identical peptides.The data from LC-MS/MS were analyzed using MaxQuant software (version 1.6.0.1; Max Planck Institute of Biochemistry, Martinsried, Germany). The combined non-redundant *Bos taurus* protein sequence database was downloaded from UniProt (www.uniprot.org (accessed on 2 March 2018). This was used for protein identification and label-free relative quantitation (LFQ). The false discovery rate was set to 0.01. Only proteins with an LFQ value of 0 and MS/MS (spectral counts) ≥2 were considered as true identification and used for further analysis.

#### 2.4.2. Sugar Analysis

The analysis for the sugar profile was conducted using gas chromatography with flame ionization detection [20,21]. This method is applicable to determine the lactose content in food samples. From each milk type, 10 g were used for the analysis. Sugar was extracted from the milk using a 50:50 methanol: water solution. The milk sample was dried with inert gas, which was derivatized prior to analysis. 

#### 2.4.3. Protein Analysis

Dumas method, also known as combustion method, was used to analyze protein in milk samples [22]. This method determines the amount of nitrogen released into pure oxygen at a high temperature. Thermal conductivity was used to measure nitrogen. The combustion instrument used was capable of analyzing ≥200 mg of the sample. The oven was operated at a temperature of ≥850 °C in pure oxygen. The high temperature in oxygen was intended to isolate nitrogen from other products of combustion (water and carbon di-oxide) and release nitrogen completely from the milk sample. Calibration of the instrument was conducted using a nitrogen standard containing 9.59% nitrogen in ethylenediaminetetraacetic acid (EDTA). The warmup time to allow the furnace and instrument to reach the operating temperature and stabilize was approximately 6 h. Milk samples were analyzed for nitrogen content following five blank and five nitrogen standard (calibration gas) analyses. The nitrogen content was then multiplied by the conversion factor (6.25) to calculate the amount of protein in the sample. 

#### 2.4.4. Fat Analysis

Base hydrolysis was the method used to determine the fat content in milk samples [23]. This method involved using concentrated ammonium hydroxide to release fat from milk. Diethyl ether and petroleum ether were used for the extraction of fat. Milk samples were then evaporated, dried and gravimetrically measured. Each milk type was tempered to 38 °C. A quantity of 10 g of milk sample was pipetted into a clean empty flask with a dry cork stopper. The first extraction was initiated by adding 1.5 mL of ammonium hydroxide to the milk sample to neutralize the acid and dissolve casein. This was followed by the addition of three drops of phenolphthalein indicator and 10 mL of 95 percent alcohol. The contents in the flask were secured with a water-soaked cork stopper and mixed for 15 s. Further, 25 mL of ethyl ether was added to the flask and vigorously mixed for one minute. Additionally, 25 mL of petroleum ether was added, and the mixture was vigorously shaken for one minute. The flask was then centrifuged at 600 rpm for ≥30 s. The ether solution was decanted and evaporated at ≤100 °C. The second extraction was done by adding 5 mL of 95 percent alcohol to the pink aqueous solution obtained after centrifugation and mixed vigorously for 15 s. A quantity of 15 mL of ethyl ether was added to the flask and mixed vigorously for 1 min. Finally, 15 mL of petroleum ether was added and mixed vigorously for 1 min. This was followed by centrifugation at 600 rpm for ≥30 s. The final extraction was done by repeating the same procedure used for the second extraction but by omitting the addition of 95 percent alcohol. The solvents were then evaporated at ≤100 °C. The extracted fat was placed in a vacuum oven at 75 °C in 20 inches of vacuum for ≥7 min and then transferred to a desiccator to cool the fat and bring it down to room temperature.

### 2.5. Study Procedure

Subjects reported to Purdue MRI Facility for two visits, at least six days apart. Subjects consumed a low-fiber dinner the night before the test feeding, fasted twelve hours prior to their visit and avoided drinking water three hours prior to the visit. On the day of the visit, scans were performed using a Magnetom 3T Prisma MRI scanner (Siemens Healthineers, Erlangen, Germany). Large flex surface coil (18-channel) and spine coil (32-channel) were used to acquire the coverage of the entire abdomen. Coronal abdominal images were acquired with HASTE sequence (repletion time (TR) = 1200 ms; echo time (TE) = 101 ms; flip angle (FA) = 160°; field-of-view (FOV) = 340 × 340 mm; in-plane resolution = 1.3 × 1.3 mm; 45 coronal slices; slice thickness = 3 mm; GRAPPA = 3). The prospective navigator-triggered HASTE sequence took about 4 min. An initial scan was performed to acquire images of the baseline empty stomach. Subjects then consumed a milk dose, and the amount of milk was calculated in ml using the following formula:$$\mathrm{Amount}\;\mathrm{of}\;\mathrm{milk}\;\mathrm{consumed}=\frac{0.5\;g\;\mathrm{of}\;\mathrm{lactose}\;\times \;\mathrm{bodyweight}\;\left(\mathrm{kg}\right)}{11\;g\;\mathrm{of}\;\mathrm{lactose}\;}\times 245\;\mathrm{mL}\;\mathrm{of}\;\mathrm{milk}\;$$

One cup of milk, or 245 mL, contains approximately 11 g of lactose. Subjects consumed 0.5 g of lactose per kg body weight. The amount of milk (mL) and lactose (g) consumed by subjects were the same during the two MRI visits. Coronal MRI images were acquired at 10, 30, 60 and 120 min with HASTE sequence after the consumption of milk. Subjects also rated and recorded their abdominal pain at 0, 30, 60, 90 and 120 min using a six-point Likert scale. The scale ranged from 0 to 5, where 0 indicated no symptom, 1 was for slight abdominal pain, 2 for mild, 3 for moderate, 4 for moderately severe and 5 for severe abdominal pain.

### 2.6. Study Ethics

The study protocol (ClinicalTrials.gov #NCT05658861) was reviewed and approved by the Purdue University Institutional Review Board (#IRB-2019-296). The study complied with the Helsinki Declaration of 1975 as revised in 2008 [24]. The trial also followed the guidelines of International Conference on Harmonization Good Clinical Practice [25].

### 2.7. Image Analysis

A software tool within FMRIB’s Software Library (FSL v5.0.11) [26] was used for volume calculation. The volume of stomach acid was calculated at baseline. A contour around the bright signal from stomach acid (Figure 2) was drawn and then auto-filled in the space within the contour to get a binary mask for each slice. Similarly, at each time point after drinking milk a contour was drawn around the milk (Figure 3); the mask was generated by auto-filling the space within the contour for each slice. The command fslstats was used to calculate the volume (mm^3^) of the mask for each time point. 

### 2.8. Statistical Analysis

The volume of both kinds of milk in the stomach during the two-hour study was compared using two-way repeated measures ANOVA. The normality of the data was tested using Shapiro–Wilk. Log transformation was used to bring the data into a normal distribution. This analysis was conducted using RStudio (v2022.07.2+576.pro12 Spotted Wakerobin, Posit, Boston, MA, USA).

The statistical difference between the volume of conventional and A2 milk at each time point was analyzed with the paired *t*-test. Bonferroni correction was used to reduce Type 1 errors. The significance level (α) was determined to be 0.0125 after the Bonferroni correction. Descriptive statistics (mean and standard deviation) were also calculated. This analysis was conducted using Microsoft^®^ Excel.

Wilcoxon signed-rank test was used to compare the symptom scores. A non-parametric test was used since the symptom scores were ordinal data rated using a Likert scale and they were not normally distributed. The analysis for abdominal pain was conducted using Statistical Package for the Social Sciences (IBM SPSS Statistics for Windows, Version 28.0; IBM Corp., Armonk, NY, USA).

Conventional milk was compared with A2 milk by measuring the volume of milk at different time points for gastric transit and with symptom scores for abdominal pain. The volume of milk was converted from millimeter cubed (mm^3^) to milliliters (ml). The gastric volume of milk was corrected for residual stomach acid by subtracting the acid volume in the stomach at baseline (before consumption of milk) from the volume of milk at each time point after the consumption of milk. The total symptom score for abdominal pain was calculated by summing all the scores over 2 h.

## 3. Results

### 3.1. Baseline Characteristics

Five out of the twenty-three subjects who were phone screened were ineligible to continue in the study due to their current medical condition, lack of milk avoidance and/or presence of metallic implants near the region of the scan. The remaining eighteen were eligible to be assessed for maldigestion with a hydrogen breath test; however, only fifteen subjects responded to attempts to schedule the screening test. Ten out of the fifteen subjects produced a 20 ppm rise in hydrogen during the six-hour milk challenge, indicating maldigestion. These ten subjects completed the two milk-feeding interventions (Figure 1) and data were analyzed for all ten participants. 

Seven female subjects and three male subjects of an age range between 19 and 42 and an average BMI of 23 kg/m^2^ completed the study. The study population included three Asians, six Caucasians and one subject that did not disclose their race. All the subjects were residing in the United States (USA) at the time of data collection. Two individuals reported Hispanic ethnicity (Table 1). 

### 3.2. Nutrient Analysis

The nutrient compositions of the A2 milk and conventional milk are shown in Table 2. All subjects consumed approximately 4.5 g of lactose/100 mL of milk. 

### 3.3. Gastric Emptying and Transit Volume

The average volume of conventional milk in the stomach over the two-hour period was significantly lower than the average volume of A2 milk (*p* < 0.001) according to two-way repeated measures ANOVA. Further, the volume of milk in the stomach was also significantly lower at 30 (*p* = 0.01), 60 (*p* = 0.002) and 120 (*p* < 0.001) minutes after ingestion of New-World milk as compared to Old-World milk (Figure 4) according to paired t-tests using a Bonferroni correction. Therefore, gastric emptying was faster when subjects consumed New-World milk as compared to A2 milk (Figure 5). 

### 3.4. Abdominal Pain

The total symptom score for abdominal pain reported by all maldigesters was 20 for conventional milk and 12 for A2 milk. This difference was not statistically significant (*p* = 0.19) (Figure 6).

### 3.5. Adverse Events

There were no adverse events reported by subjects during the study. 

## 4. Discussion

No previously reported studies have compared the gastric emptying time of milk containing A1 and A2 β-casein with milk containing only A2 β-casein. Therefore, this pilot study could be used for power calculations for larger-scale studies. We planned to recruit 10 subjects to examine a difference in transit. We completed the trial after we collected data from all 10 participants. While the sample size is modest, the differences in gastric transit are highly significant and biologically relevant. 

We suggest that the gastric transit of New-World milk containing both A1 and A2 β-casein is more rapid as compared to Old-World milk containing only A2 β-casein. Jianqin et. al. reported a slower total gastrointestinal and colonic transit of milk containing A1 β-casein in comparison to A2 milk [11]. Faster transit through the stomach may result in less small intestinal digestion due to an overwhelming of the mucosal lactase by lactose. Rapid transit may deliver more unabsorbed carbohydrates into the colon [27]. The incompletely digested milk likely takes a longer time to transit the colon, resulting in more hydrogen, methane and carbon dioxide, and more discomfort [28]. Furthermore, A1 β-casein induced a pro-inflammatory effect in the colon [29]. Transit and inflammation were mediated by opioid receptors in rodents [22]. BCM-7 is a pro-inflammatory opioid peptide [30] and has an affinity to the micro-opioid receptors in the gastrointestinal tract of humans [31,32,33]. Casomorphin slows the transit by decreasing gastrointestinal motility, which is similar to the effect of an opioid and opioid agonist [34,35,36]. However, the result from our study suggests that A1 β-casein has the opposite effect in the stomach.

Slow gastric emptying is associated with better tolerance and digestion among maldigesters [37]. Since delaying gastric transit of food considerably reduces abdominal pain in lactose maldigesters [28], we expected a reduction in abdominal pain when subjects consumed A2 milk that transited slower from the stomach than conventional milk. Although the symptom score for abdominal pain was lower with A2 milk, the difference was not significant, probably due to the limited power of the sample size. However, our primary endpoint of gastric transit showed remarkable differences. Consumption of A2 milk resulted in a gastric volume that was approximately 2.5 times larger at 120 min as compared to the consumption of conventional milk. This very large difference in volume is likely attributed to the β-casein variant. The total gastrointestinal transit time of A1 β-casein was longer than A2 β-casein in rats [29] and humans [11]. 

We did not stratify subjects based on age, gender or race because evidence suggests that lactose intolerance does not depend on demographics but on the dose and differences in the lactase non-persistent gene [38]. However, it was found that the volume of conventional milk was lower than the volume of A2 milk in the stomach at the end of the two-hour study, or the 120-min timepoint, for all 10 subjects, indicating that the transit of conventional milk is more rapid than A2 milk, regardless of demographics. Future studies with larger populations could validate these findings.

The nutrient composition of conventional milk and A2 milk were very similar. The slight difference in fat could account for some of the difference in transit [39]. The amount of fat in A2 milk was 10% more than in conventional milk. Low-fat kinds of milk vary in nutrient composition, dairy, herd type and season. Food and Drug Administration (FDA) regulations for low-fat milk indicate a range of fat content from 0.5% to 2% [40]. United States Department of Agriculture (USDA) food database reports a variation of 5% for fat content and 16% for calorie content between 2% reduced fat conventional milk and 2% reduced fat A2 milk [41,42]. Formulating milk varying only in the β-casein variants using separation technologies to produce milk with identical lipid, carbohydrate and protein content, and examining gastric transit could resolve this question. However, from a practical perspective, our findings from this study and prior work showing more abdominal discomfort with milk containing A1 β-casein suggest that consumer choice of low-fat milk with only A2 β-casein would likely result in a dramatic difference in gastric emptying and some difference in intolerance symptoms [13].

## 5. Conclusions

In summary, the gastric transit of milk containing both A1 and A2 β-casein was faster as compared to milk containing only A2 β-casein. The abdominal pain resulting from New-World milk and A2 milk was not different, likely due to the small sample size. The transit findings reported here could be the reason for less maldigestion following the consumption of milk containing only A2 β-casein as compared to milk containing both A1 and A2 β-casein in lactose maldigesters [13,14]. Studies utilizing a larger sample size and eliminating the confounding effect of other nutrients in milk would sort out the potential effects of composition and A1 β-casein on gastric emptying. 

## Figures and Tables

**Figure 1 nutrients-15-00801-f001:**
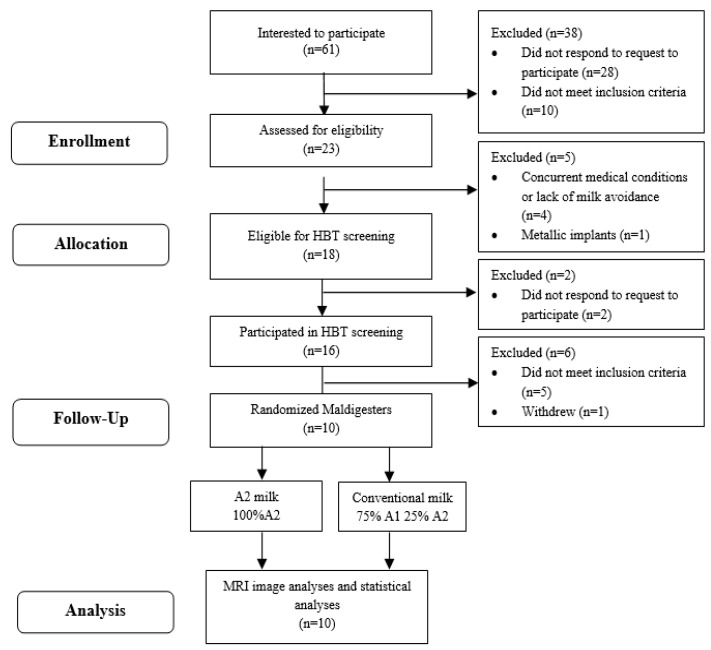
Study flow. HBT, hydrogen breath test; MRI, magnetic resonance imaging analysis.

**Figure 2 nutrients-15-00801-f002:**
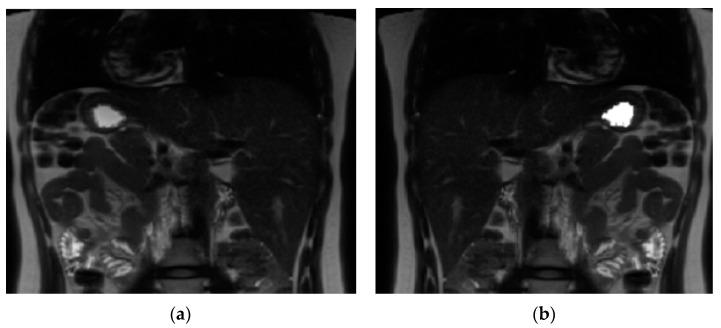
Image analysis using FMRIB’s Software Library (FSL v5.0.11). (**a**) Magnetic resonance image of the stomach before drinking milk. (**b**) Magnetic resonance image of the stomach with the mask of the stomach acid content at baseline.

**Figure 3 nutrients-15-00801-f003:**
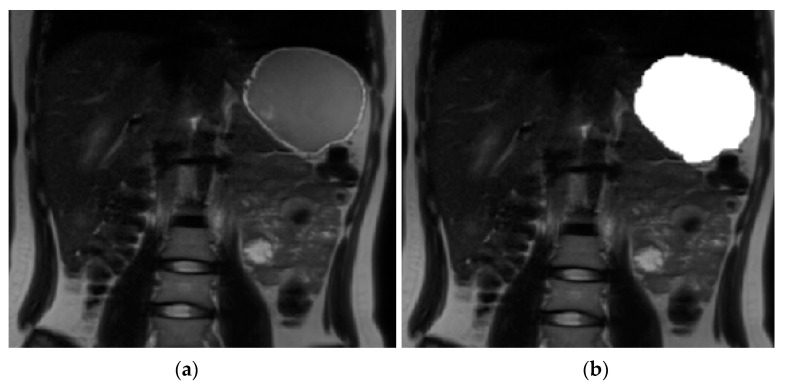
Image analysis using FMRIB’s Software Library (FSL v5.0.11). (**a**) Magnetic resonance image of the stomach 10 min after drinking milk. (**b**) Magnetic resonance image of the stomach with the mask of the milk content.

**Figure 4 nutrients-15-00801-f004:**
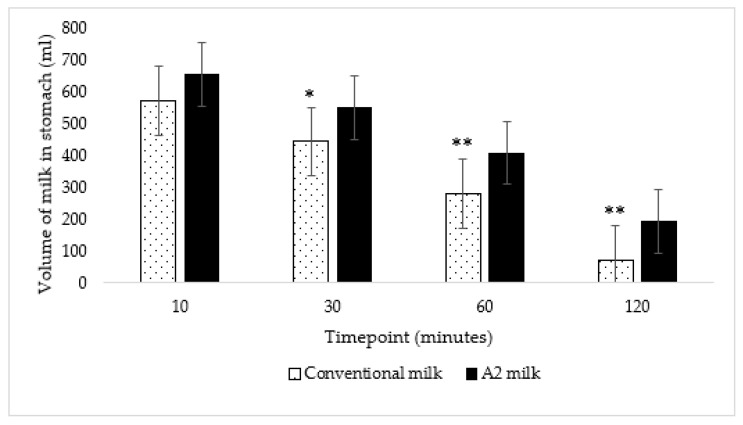
The volume of A2 milk in the stomach was significantly lower at 30, 60 and 120 min as compared to the volume of conventional milk in the stomach: * *p* = 0.01 at 30 min and ** *p* < 0.01 at 60 and 120 min; *p* = 0.05, *p* = 0.01, *p* = 0.002, and *p* < 0.001 at 10, 30, 60 and 120 min, respectively. The vertical lines in the graph indicate the standard error.

**Figure 5 nutrients-15-00801-f005:**
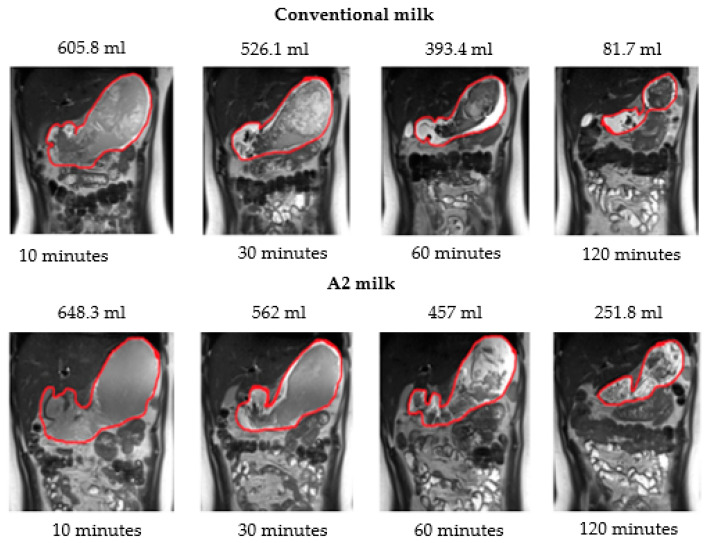
Magnetic resonance images showing the gastric volume of conventional milk and A2 milk and volume calculated in milliliters (mL) at 10, 30, 60 and 120 min.

**Figure 6 nutrients-15-00801-f006:**
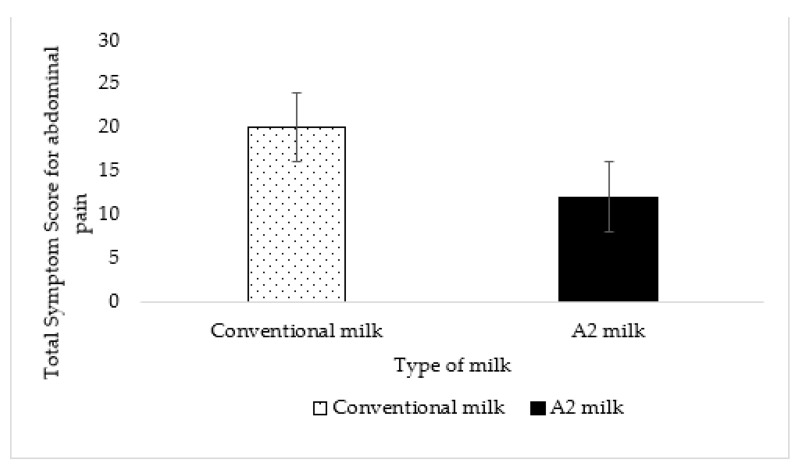
No effect or difference was detected in abdominal pain between A2 milk and conventional milk (*p* = 0.19). The vertical lines in the graph indicate the standard error.

**Table 1 nutrients-15-00801-t001:** Baseline and demographic characteristics of lactose maldigesters enrolled in the trial.

Age, mean (range); years	27 (19–42)
Bodyweight, mean (range); kg	62 (40–91)
Height, mean (range); cm	164 (150–185)
BMI, mean (range); kg/m^2^	23 (18–27)
Male/female, n/n	3/7
Asian	3
Caucasian	6
Unknown or not reported	1
Hispanic	2
Non-Hispanic	8

**Table 2 nutrients-15-00801-t002:** Nutrient composition of the two milk treatments.

Nutrient	A2 Milk or Old-World Milk	Conventional Milk or New-World Milk
Protein (g/serving)	3.14	3.30
Fat (g/serving)	2.10	1.90
Lactose (g/serving)	4.70	4.60
Carbohydrate (g/serving)	4.70	4.60
Calories (kcal/serving)	0.0541	0.0500
A1 β-casein protein (%)	0.00	75.00
A2 β-casein protein (%)	100.00	25.00

## Data Availability

The data presented in this study are available on request from the corresponding author. The data are not publicly available due to privacy and ethical restrictions.

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
