# Peer review of "Gastric Emptying of New-World Milk Containing A1 and A2 Β-Casein Is More Rapid as Compared to Milk Containing Only A2 Β-Casein in Lactose Maldigesters: A Randomized, Cross-Over Trial Using Magnetic Resonance Imaging"

_nutrients, 2023, doi:10.3390/nu15040801_

Round 1

Reviewer 1 Report

1.     Abstract – Line 16-17 The two “or” in the sentence may confuse the reader who only read the Abstract. Please consider revising it to make it clearer.

2.     Figure 3 – study flow should be moved to close to Method (section 2.1).

3.     Why did the authors use paired t-tests for statistical analysis but not repeated measures of ANOVA? It is also not appropriate to use t-tests for non-continuous data like the abdominal pain scale.

4.     Line 241 – what endpoint measurements in this study could indicate less maldigestion?

Author Response

Thank you for your feedback. Please see the attachment with responses to the comments.

Reviewer 2 Report

The authors have conducted an interesting study on the relationship between the content of A1 casein in milk and the gastric emptying time in lactose maldigesters. Please consider the following notes:

Comment 1: When responding each comment, please indicate the exact location in the new version of the manuscript. 

Comment 2: Please clarify better the terminology used for old world and new world milks. Do new and old refer to the common meaning of new and old words? Please be explicit in the introduction. Also, please standardize their writing: capital initials or not; hyphened or not.

Comment 3: Lines 73, 74 should be in past tense. Please also double-check the whole manuscript for the correct tense.  

Comment 4: Please add an appropriate reference in line 79 for HBT.

Comment 5: The detailed composition of both milk sources must be shown in the manuscript. Referring Ramakrishnan et al (2020) is not enough, as it has to be explicitly evidenced that the effects is likely caused by the content of casein A1, not due to another difference.

Comment 6: Please clarify how did you come up with the coefficients in the equation in line 107 (11 g lactose; 245 ml milk).

Comment 7: Please add an appropriate reference in line 113 for the Helsinki declaration.

Comment 8: Please add an appropriate reference in line 114 for the mentioned guidelines.

Comment 9: In line 122, please indicate the units for volume.

Comment 10: Please check the syntax of the sentence in lines 133-134. The beginning has a problem ("this is study that").

Comment 11: Please clearly describe before the section Statistical Analysis the procedure to determine the symptom score, including the units.

Comment 12: Scores (symptom score) should be analyzed with a non-parametric test due to a lack of normal distribution. Please re-analyze this variable with an appropriate test or clearly justify the parametric test you used.

Comment 13: If you corrected by running a non-parametric test, please ensure everything relying on that is updated (e.g., results, discussion).

Comment 14: Please clarify the procedure to correct for residual stomach acid (line 144).

Comment 15: Standardize the number of decimal digits for P values. No more than 4 digits, please.

Comment 16: In figures 4 and 6, please indicate in the caption the meaning of the vertical lines crossing the bars (e.g., standard error, standard deviation).

Comment 17: The sentence in lines 215-216 is fallacious and should be removed. In fact, you are using that sample size to build inferences.

Comment 18: The inference in line 219 ("is likely attributed to...") cannot be supported if the detailed composition of the milk sources used is not shown in the manuscript.

Author Response

(The authors gave the same response as above.)

Reviewer 3 Report

The work analizza the consumption of milk containing a mixture of A1 and A2 β-casein as compared to milk containing only A2 β-casein: the Gastric transit affects gastrointestinal symptoms and rapid transit has been associated with an

increase in abdominal pain.

In particular, the two variants of lactose in gastric and intestinal transit are analyzed in order to evaluate a direct correlation between rapid and slow gastric emptying with NMR and subsequent breakdown of lactose.

Although the question of demonstrating the more digestible variant of the less digestible one has been demonstrated, it is not supported by a large case study that could reconfirm the results.

Therefore the data are encouraging but as a premise of a work that will have to be articulated on a wide range of cases

Round 2

Reviewer 2 Report

Thank you for taking into account previous comments.

In the captions of Figure 6. You may consider that a P>0.05 does not prove treatments are the same; it simply indicates the null hypothesis was not rejected, so it's not possible to state that they are different. This means the most appropriate interpretation for P>0.05 is that "no effect or difference was detected" or anything equivalent. For a very well-informed reader "was not different" or "showed no effect" will mean "no difference or effect was detected". For a not informed enough reader it will mean "no effect exists"; then they may talk about "contradictory results". You may consider this criterion to rephrase your inferences and support readers' interpretation. Indeed, it is possible that the number of available subjects in the study was not enough to get enough statistical power. This applies to line 237 and everywhere else appropriate.

Author Response

Thanks for the comment. 

Lines 237 to 238: Changed the sentence to "No effect or difference was detected in abdominal pain between A2 milk and conventional milk (p=0.19)"